DOI: 10.1038/s41467-018-05114-7　　**OPEN**

# Population genomics of hypervirulent *Klebsiella pneumoniae* clonal-group 23 reveals early emergence and rapid global dissemination

Margaret M. C. Lam[1], Kelly L. Wyres[1], Sebastian Duchêne[1], Ryan R. Wick[1], Louise M. Judd[1], Yunn-Hwen Gan[2], Chu-Han Hoh[2], Sophia Archuleta[3], James S. Molton [3], Shirin Kalimuddin[4], Tse Hsien Koh[5], Virginie Passet[6], Sylvain Brisse[6] & Kathryn E. Holt [1,7]

Severe liver abscess infections caused by hypervirulent clonal-group CG23 *Klebsiella pneumoniae* have been increasingly reported since the mid-1980s. Strains typically possess several virulence factors including an integrative, conjugative element ICE*Kp* encoding the siderophore yersiniabactin and genotoxin colibactin. Here we investigate CG23's evolutionary history, showing several deep-branching sublineages associated with distinct ICE*Kp* acquisitions. Over 80% of liver abscess isolates belong to sublineage CG23-I, which emerged in ~1928 following acquisition of ICE*Kp10* (encoding yersiniabactin and colibactin), and then disseminated globally within the human population. CG23-I's distinguishing feature is the colibactin synthesis locus, which reportedly promotes gut colonisation and metastatic infection in murine models. These data show circulation of CG23 *K. pneumoniae* decades before the liver abscess epidemic was first recognised, and provide a framework for future epidemiological and experimental studies of hypervirulent *K. pneumoniae*. To support such studies we present an open access, completely sequenced CG23-I human liver abscess isolate, SGH10.

[1] Department of Biochemistry and Molecular Biology, Bio21 Molecular Science and Biotechnology Institute, University of Melbourne, Parkville, Victoria 3010, Australia. [2] Department of Biochemistry, Yong Loo Lin School of Medicine, National University of Singapore, Singapore 119228, Singapore. [3] Department of Medicine, Yong Loo Lin School of Medicine, National University of Singapore, Singapore 119228, Singapore. [4] Department of Infectious Diseases, Singapore General Hospital, Singapore 169608, Singapore. [5] Department of Microbiology, Singapore General Hospital, Singapore 169608, Singapore. [6] Institut Pasteur, Biodiversity and Epidemiology of Bacterial Pathogens, 75015 Paris, France. [7] The London School of Hygiene and Tropical Medicine, London WC1E 7HT, United Kingdom. These authors contributed equally: Margaret M. C. Lam and Kelly L. Wyres. Correspondence and requests for materials should be addressed to K.E.H. (email: kholt@unimelb.edu.au)

**K**lebsiella pneumoniae (Kp) is a ubiquitous bacterium and important cause of multidrug-resistant healthcare-associated infections. The past three decades have also seen the emergence of severe community-acquired hypervirulent Kp disease, usually manifesting as pyogenic liver abscess with accompanying bacteraemia, but also meningitis, brain abscess or opthalmitis[1,2]. Earliest reports of this syndrome emerged across parts of Asia including Taiwan, China, Hong-Kong, Singapore and South Korea and more recently from Europe, the United States, South America, the Middle East, and Australia[1,2].

The majority of Kp liver abscess isolates belong to a small number of clonal groups (CGs)[3–5] defined by multi-locus sequence typing (MLST)[6]. In particular, CG23 was shown to account for 37–64% isolates in Taiwan[5], Singapore[7] and mainland China[4,8], and includes sequence types ST23, ST26, ST57 and ST163[3]. CG23 Kp are associated with the highly serum-resistant K1 capsule and a number of virulence factors; the genotoxin colibactin, microcin E492, and the iron-scavenging siderophores aerobactin, yersiniabactin and salmochelin[1,9,10]. The siderophores are associated with the ability to cause disseminated infection in mouse models[10–13] and with invasive infections in humans[9,13,14].

Two CG23 comparative genomic analyses have been published, incorporating up to 27 isolates[15,16]. Both studies reported limited nucleotide variation in core chromosomal genes and high conservation of the acquired virulence genes. These include the K1 capsule synthesis locus KL1, the chromosomally encoded yersiniabactin locus (ybt), plus the iro (salmochelin), iuc (aerobactin) and rmpA/rmpA2 genes (upregulators of capsule expression) located on the pK2044 virulence plasmid[17]. The ybt locus is mobilised by the ICEKp integrative conjugative element[18], for which we have recently described over a dozen variants in the wider Kp population[14]. Variants harbouring ybt, plus iro (ICEKp1) or the colibactin locus (clb, ICEKp10) have been reported in CG23[14,18,19]. ICEKp1 was the first ICE to be described in Kp[18], originating from the sequence type (ST) 23 liver abscess strain NTUH-K2044, which was also the first CG23 strain to be completely sequenced[17]. However ICEKp10 is more common and was observed in all other CG23 genomes investigated in Struve et al[16].

CG23 has not generally been associated with acquired antimicrobial resistance (AMR), but the last few years have seen increasing reports of resistant strains, including those resistant to third generation cephalosporins and carbapenems[15,20–22]. The potent combination of virulence and AMR determinants could make these strains a substantial public health threat, but it is not yet clear how often they emerge or whether they can disseminate. To fully evaluate the threat we require a clear understanding of the history of this clone.

Here we describe an updated evolutionary history for CG23 based on genomic analysis of 98 human and equine associated isolates, representing the largest and most geographically diverse collection to date. Our data reveal previously undetected population structure, provide sufficient temporal signal to estimate the date of emergence of CG23, and indicate that the commonly used NTUH-K2044 reference strain is not representative of a 'typical' CG23.

## Results

**Phylogenomics and evolutionary history of CG23.** Comparative analysis of the 97 CG23 genomes (Supplementary Data 1) identified no recombination events (besides the known hybrid strain CAS686[16] which was excluded from analysis, see below), and showed low nucleotide divergence across core chromosomal genes, with a median pairwise distance of 233 single nucleotide polymorphisms (SNPs; range 1–444 SNPs), and 0.0045% nucleotide divergence (range 0.000019–0.0086%).

Phylogenetic analyses indicated that CG23 has a number of sublineages separated by deep-branches, one of which (labelled CG23 sublineage I, CG23-I) has become globally distributed (Fig. 1, Supplementary Fig. 1). The CG23-I clade was strongly supported by two independent analysis methods (>99% posterior support in Bayesian tree, 100% bootstrap support in ML tree). This clade comprised 81 isolates (83.5% of all CG23) collected from Asia, Australia, North America, Europe and Africa (Fig. 1), including 82% of all liver abscess strains (Supplementary Data 1). Neither of the two oldest strains in our analysis, M109 (1932, Murray Collection[23]) and NCTC9494 (1954, human sputum), were part of CG23-I (Fig. 1); nor was NTUH-K2044, the first sequenced ST23 strain that has served as a reference for much of the reported experimental and genomic work on ST23[16–18].

CG23-I was separated from the rest of CG23 by 49 SNPs, including one nonsense mutation (Supplementary Data 2). The nonsense mutation is predicted to truncate the outer membrane usher domain of KpcC, likely preventing expression of the Kpc fimbriae, which were discovered in the NTUH-K2044 genome and have been proposed to be characteristic of K1 liver abscess strains[16,24]. Genes with non-synonymous mutations include those encoding two putative efflux pumps, the AmpC beta-lactamase, the PmrB sensor kinase, four membrane transport/secretion proteins, an acid shock protein and the cardiolipin synthase 2 protein (Supplementary Data 2).

Kp is considered an important cause of sexually transmitted disease in horses[25] but little is known about the molecular epidemiology of this group or sources of infection. The horse isolates included in this study (including genital tract, sperm, foetus and metritis specimens isolated between 1980 and 2004) were originally selected for sequencing only on the basis of K1 serotype, but clustered together within a monophyletic subclade nested within CG23-I (Fig. 1), separated from the rest of CG23-I by 83 SNPs (Supplementary Data 3). Of note, four non-synonymous mutations arose within genes encoding putative oxidoreductases and two intergenic mutations were within close proximity to predicted oxidoreductase genes.

The nested positioning within CG23-I suggests that CG23 Kp may have entered the horse population on a single occasion where it now circulates via sexual contact, and there is no evidence of transmission between humans and horses.

We detected a strong temporal signal in the CG23 genome alignment (see Supplementary Fig. 2a-c, Supplementary Methods), sufficient to estimate evolutionary rates and dates for the most recent common ancestors (MRCAs) of key CG23 lineages with BEAST (best-fitting Bayesian model was the UCLD constant model, see Supplementary Methods). The mean evolutionary rate for CG23 was estimated to be $3.40 \times 10^{-7}$ substitutions site$^{-1}$ year$^{-1}$ (95% HPD; $2.43 \times 10^{-7}$–$4.38 \times 10^{-7}$). The MRCAs for the entire CG23 population, CG23-I and equine sublineage nested within CG23-I were estimated to be 1878 (95% HPD; 1827–1915), 1928 (95% HPD; 1908–1953) and 1972 (95% HPD, 1961–1975), respectively. We hypothesised that the successful spread of CG23-I in the human population may have been associated with a population expansion; i.e. an increase in diversification rate and effective population size, compared to the rest of the CG23 population. We used an epidemiological model (birth-death) to compare these dynamics between the different lineages (see Supplementary Methods), which demonstrated population decline in the horse lineage and expansion of the entire human-associated CG23 population (Supplementary Fig. 3a, c), particularly among CG23-I, which was associated with five times the rate of expansion than the rest of the population (Supplementary Fig. 3b, d). Importantly, there was

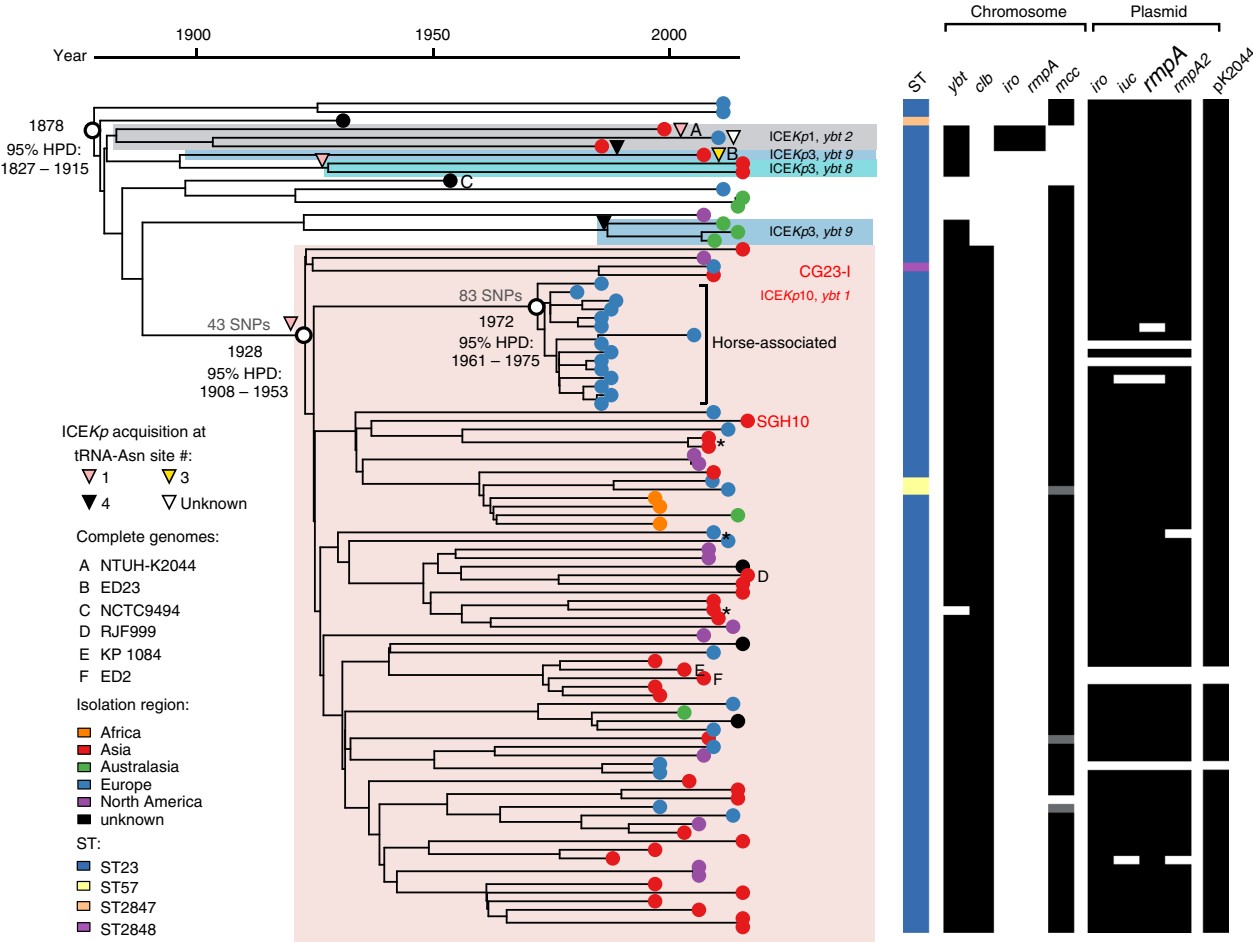

**Fig. 1** Phylogenetic relationships within CG23. Time-calibrated Bayesian phylogeny (left) showing the relationships between 97 CG23 *Kp*, their sequences types (STs), the presence of chromosomal virulence loci (*ybt*, yersiniabactin; *clb*, colibactin; *iro*, salmochelin; *rmpA*, regulator of mucoid phenotype; *mcc*, microcin E492), plasmid virulence loci (*iro* salmochelin, *iuc* aerobactin, *rmpA/rmpA2* regulator of mucoid phenotype), and the virulence plasmid backbone (pK2044-like, see Supplementary Fig. 6). Grey shading indicate partial *mcc* deletions (see Supplementary Fig. 7). Tips are coloured by region of isolate origin as indicated. In addition to SGH10, complete genomes for which chromosomes are shown in Fig. 2 are marked A–F as per inset legend. * indicates three human-associated multidrug resistant isolates. Subclades containing ICE*Kp* are indicated by shading, labelled with the corresponding ICE*Kp* and *ybt* lineages; the pink shaded clade carrying ICE*Kp10* (*ybt* 1) is the dominant lineage, CG23-I. Putative ICE*Kp* acquisition events by a lineage or single strain are indicated by triangles at the corresponding node or next to a strain, and are shaded by chromosomal tRNA-Asn insertion site. Bayesian molecular dating estimates are shown for the most recent common ancestor of all isolates (i.e. the tree root), the CG23-I lineage, and the horse lineage (nodes as marked with white circles). The number of SNPs unique to CG23-I and the horse lineage are indicated on the relevant branches

significant overlap in the Bayesian parameter estimate distributions from four alternative models, and with estimates derived from an alternative method known as least-squares LSD analysis (Supplementary Fig. 2d-f, see Supplementary Methods). This provides confidence that CG23 emerged some time in the late 19th century and the globally distributed CG23-I emerged in the 1920s.

**Isolate CAS686 is a hybrid of ST23 and ST281-like *Kp*.** The genome of isolate CAS686 (ST260, a double locus variant of ST23) was previously shown to contain a large recombination import spanning approximately half of the genome (2.4Mbp)[16]. Here, we identified the likely hybridisation partner to be a ST281-related *Kp*. We used the RedDog mapping and variant calling pipeline to compare the genome of CAS686 to those of CG23 and other genomes in our curated collection[14,26]. The distribution of SNPs differentiating CAS686 from other CG23 genomes confirms the previous observation that approximately half of the CAS686 genome is closely related to ST23 (Supplementary Fig. 4a). However, the region of similarity appears to be split into two,

separated by regions of similarity to the genomes of QMP M1-975 (isolated from asymptomatic human gut carriage[9]) and INF211 (isolated from a case of pneumonia in an Australian hospital[27]), both of which are double locus variants of ST281 (Supplementary Fig. 4b).

A subsequent Bayesian phylogenetic analysis showed that the ST23-like regions of CAS686 cluster within CG23-I (models and parameters as described for the analysis presented above, all CG23 genomes were included and the recombination regions in CAS686 were masked). However, it is not possible to know whether CAS686 represents a ST23 strain that has acquired two large regions from an ST281-related strain, or vice versa. Note though that CAS686, but neither QMP M1-975 nor INF211, harbours the virulence plasmid encoding *iro*, *iuc* and *rmpA/A2* loci that is characteristic of CG23, indicating that import into CG23 is most parsimonious.

**Virulence gene content variation in CG23.** To investigate the ongoing evolution of virulence in CG23, we performed detailed analyses of variation in virulence gene content within the

population. All CG23 genomes (including the hybrid strain CAS686) carried the KL1 locus associated with biosynthesis of the K1 capsule serotype. The K-locus of the 1954 isolate NCTC9494 harboured a 1065 bp insertion between the *wzi* and *wza* genes, which contained an IS*102*-like transposase; insertions in K-loci are common amongst historical isolates[26] and likely arise during long-term storage.

The *ybt* locus was present in 90 CG23 genomes (93%). Comparison of *ybt* lineages, ICE*Kp* structures and chromosomal integration sites indicated at least six independent *ybt* + ICE*Kp* acquisitions (triangles in Fig. 1). CG23-I was characterised by the presence of ICE*Kp10* encoding *ybt* lineage 1 (*ybt*-1) and *clb* lineage 2 A sequence variants, integrated at tRNA-Asn site 1, consistent with a single integration event in the ancestor of the sublineage. One isolate from this group lacked the *ybt* genes (Fig. 1) but harboured the left and right ends of ICE*Kp10* (Supplementary Fig. 5), consistent with this genome sharing the historical acquisition of ICE*Kp10* followed by subsequent loss of *ybt* and the ICE*Kp* mobilisation machinery.

At least five unrelated ICE*Kp* integration events were identified outside of CG23-I (Fig. 1), none of which included *clb*. Three separate introductions of ICE*Kp3* were detected: ED23 (from Taiwan) carried *ybt*-9 integrated at tRNA-Asn site 3, three isolates from Australia carried *ybt*-9 variant integrated at tRNA-Asn site 4, and two related Singapore isolates carried *ybt*-8 integrated at tRNA-Asn site 1. ICE*Kp1* encoding *ybt*-2 was identified in two strains from Taiwan (including the NTUH-K2044 reference strain) and one strain from France. These strains were monophyletic in the tree (grey shaded clade in Fig. 1) and share a MRCA close to the root, however the chromosomal locations of ICE*Kp1* suggest they may have been acquired via distinct integration events in each strain: at site 1 in NTUH-K2044 and site 4 in SB3926 (the integration site could not be resolved for the French isolate SB4446).

The virulence plasmid-associated loci *iuc*, *iro*, *rmpA* and *rmpA2* were present in the vast majority of CG23 genomes (see Fig. 1). Comparison of genome sequences to the reference sequence of virulence plasmid pK2044 (strain NTUH-K2044) confirmed that the plasmid backbone was present in 94 genomes, including all those carrying *iuc*, *iro*, *rmpA* and/or *rmpA2*, although some plasmids harboured deletions that affected these virulence loci (see Fig. 1, Supplementary Fig. 6a; note it is unclear whether the three plasmid-negative isolates were actually lacking the plasmid in vivo or had lost it during laboratory culture). Other virulence loci detected in CG23 included microcin E492 (only three deletion variants detected, see Supplementary Fig. 7) and allantoinase, both of which were found in the chromosome.

The microcin E492 ICE was present in 89 of the CG23 genomes (92%, Fig. 1), integrated at tRNA-Asn site 2 in all cases (except four genomes in which the site could not be resolved from the available short-read data, see Supplementary Data 1). In the genomes lacking microcin, the tRNA-Asn 2 region was intact and showed no other integration at this site. Sequence homology and structure of the microcin locus was generally conserved, with no more than four SNPs between pairs of microcin ICE sequences. However three strains carried putative deletion variants (Supplementary Fig. 7), consistent with three independent deletion events (grey in Fig. 1 heatmap). The chromosomal allantoinase locus was present in all CG23 genomes, consistent with previous reports[15,16].

**Antimicrobial resistance determinants in CG23.** All CG23 genomes carried the intrinsic beta-lactamase gene *bla*$_{SHV}$ (which confers ampicillin resistance) and the *oqxAB* genes (which confer reduced susceptibility to quinolones). However no fluoroquinolone

resistance-associated mutations were identified in *gyrA* or *parC*, and acquired AMR determinants were rare. The equine clade and three other unrelated CG23-I isolated from humans (CAS813, BG130 and BG141; * in Fig. 1) carried multiple AMR genes.

The equine clade was associated with *aadA1* and *aph3"Ia* (aminoglycosides), *sul1* (sulphonamides) and *tetAR* (tetracycline), carried in an IncFII plasmid backbone (see Supplementary Fig. 8). We could not resolve complete resistance regions from all the short-read assemblies, but we were able to extract a putative plasmid sequence from the assembly graph of isolate SB4816, which was subsequently resolved into the complete plasmid sequence pBSB4816 using long read sequencing (deposited under GenBank accession 'MF363048', see Methods). The pBSB4816 sequence comprised an IncFII type plasmid backbone harbouring a Tn*21*-like element that carried a class I integron with *aadA1* in the gene cassette and *sul1* downstream of the *qacE*delta1 fragment (similar to GenBank accession: 'AF071413'[28], see Supplementary Fig. 8a). The Tn*21*-like element was interrupted by IS*26* sequences inserted in the *tnpR* and *tniA* genes. The plasmid backbone showed close similarity (99% nucleotide identity) to plasmid pEC_L8 from a clinical *E. coli* isolate[29]. Mapping of reads to the plasmid sequence from SB4816 confirmed that a variant of the plasmid was present in all of the AMR equine clade genomes and absent from the rest of the CG23 population (Supplementary Fig. 6). Where *aph3"-Ia* or *tetAR* were present they were also located in sequences flanked by IS*26* (see Supplementary Fig. 8b), hence we propose that variation in AMR gene content within the equine clade is due to IS*26*-mediated mobilisation or deletion of these AMR genes.

The multidrug-resistant strains isolated from humans each harboured unique sets of AMR genes conferring resistance to multiple classes of drugs, as previously reported[15,16] (see Supplementary Data 1). While precise locations of the AMR genes were not resolvable in the public draft genome assemblies, all three were found to harbour large conjugative plasmids that are frequently associated with multidrug resistance in Enterobacteriaceae: IncA/C$_2$ plasmid sequence type (PST) 3 in CAS813 (Denmark, 2008) and BG130 (Vietnam, 2008); IncN PST6 in BG141 (Madagascar, 2007).

**Other gene content variation in CG23.** We compared the seven completely assembled CG23 chromosomes, comprising SGH10 plus five clinical *Kp* isolates from Taiwan and China collected between 2002 and 2015 (NTUH-K2044;[17] 1084;[30] RJF999; ED2; ED23) and NCTC9494 from the Public Health England National Culture Type Collection (PHE NCTC, collected in 1954 and recently sequenced as part of the NCTC3000 genomes project, see http://www.sanger.ac.uk/resources/downloads/bacteria/nctc/).

The analysis revealed remarkable conservation of gene content and synteny (Fig. 2). Relative to the majority consensus, chromosomal inversions of ~1.45 Mbp and ~472 kbp regions were observed in strains ED2 and RJF999, respectively (Fig. 2). The boundaries of the RJF999 inversion were flanked by copies of the transposase ISKpn1. Phage sequences flanked the inverted region in ED2. Variations in gene content were mainly due to integrations of prophage, ICE*Kp* variants and the microcin ICE.

Pan-genome analysis of the full set of 97 genomes identified 4170 core genes (present in ≥95% of genomes) and 5493 accessory genes (see Supplementary Fig. 9). Nineteen coding sequences were uniquely present in CG23-I and conserved within this lineage (>95% CG23-I genomes). Eighteen of these were part of the colibactin synthesis locus *clb* (detailed below) and the other was an ISKpn1 transposase inserted within an ethanolamine transporter gene, *eat* (KP1_1165 in NTUH-K2044).

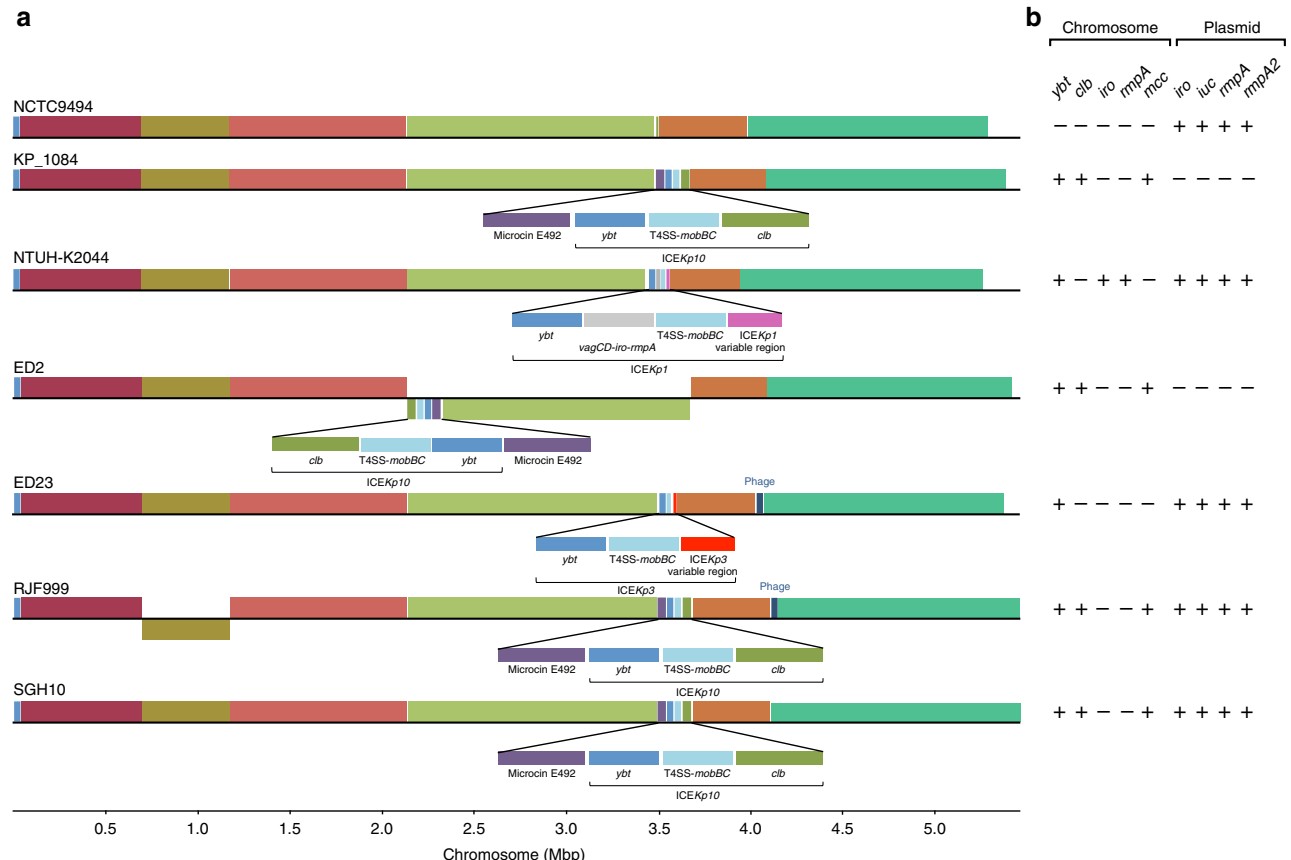

**Fig. 2** Chromosomal synteny and content comparisons between CG23 completed assemblies. **a** Homologous regions common to the chromosomes for six completed CG23 chromosomes are represented as blocks. Chromosomal inversions relative to the oldest genome NCTC9494 are indicated by blocks below the line, while blocks above the line indicate the same orientation as the reference. Unique acquisitions that were not common to all genomes are annotated. **b** Presence (+) or absence (−) of key chromosomal- and plasmid-encoded virulence loci in completed CG23 genomes

To estimate the extent to which phage account for differences in gene content, we submitted all 97 assemblies for PHAST[31] analysis. The number of complete phage sequences detected (listed in Supplementary Data 1) ranged from 0 to 3 per genome (mean = 1.4), and the sizes of these phage varied from 20 kbp to >100 kbp. These phage sequences account for 1238 (12.8%) of the coding sequences in the pan-genome (BLASTn coverage ≥90%, identity ≥73.8%, coloured red in Supplementary Fig. 9). However, this is likely an underestimate given that putative partial phage sequences were excluded from this comparison because they cannot be completely resolved from draft genome assemblies. The remaining rare coding sequences were likely associated with plasmids: we detected 10 distinct plasmid *rep* genes (Supplementary Data 1), including those associated with incompatibility types FIBk and HI1B (virulence plasmid); the FII, N and A/C2 plasmids noted above in AMR isolates; and five other F type plasmid replicons (FIA, FIIk, FIB, FIBMar, FIBpKPHS1). The gene content matrix (Supplementary Fig. 9) reveals very few examples of groups of accessory genes that were shared between multiple strains, most of which were associated with phage or with the AMR plasmid conserved in the equine strains (coloured in blue in Supplementary Fig. 9).

Most CG23 genomes (n = 93, 94.9%) harboured a complete set of type I-E CRISPR/Cas systems comprising all seven *cas* genes and two CRISPR arrays in line with previous analyses of strains NTUH-K2044 and 1084[32,33] (Supplementary Data 1). The system in NTUH-K2044 has been shown to be active, causing a reduction in the rate of in vitro plasmid acquisition and stability[33]. Here we identified a total of 259 unique spacer sequences amongst CG23 genomes. The median number of spacers per genome was 22 (range 11–55 amongst strains with intact *cas* loci), with a median of seven spacers shared with NTUH-K2044, suggesting the CRISPR/Cas system is also active in the majority of CG23 *Kp*, which may explain the relatively low level of horizontal gene transfer (including plasmid burden[34] and recombination[35]) detected in this clone compared to other clonal groups such as CG258[36].

**SGH10 as a novel reference strain for hypervirulent CG23.** Our data on CG23 population structure and virulence loci show that none of the currently available finished genome sequences represent a 'typical' genome (Figs. 1 and 2b). NTUH-K2044 (liver abscess), ED23 (blood) and NCTC 9494 (sputum) are not part of the predominant CG23-I sublineage and do not harbour the colibactin locus (labelled A-C in Fig. 1; note that NTUH-K2044 also carries additional chromosomal copies of *iro* and *rmpA* within ICE*Kp1* that are absent from the majority of CG23). Strains 1084 (liver abscess) and ED2 (blood) belong to CG23-I but lack the virulence plasmid, and RJF999 is a blood isolate of undetermined virulence. We therefore propose strain SGH10 as a reference strain for experimental and genomic studies of hyper-virulent CG23 *K. pneumoniae* associated with human liver abscess because it belongs to the predominant CG23-I sublineage responsible for the majority of CG23-associated liver abscess infections, and it has all of the common virulence loci intact with no atypical accessory genes (see Fig. 1 and Supplementary Fig. 9). SGH10 was isolated from a 35-year-old liver abscess patient with

no diabetes and no detectable underlying disease, in Singapore in 2014. We have previously demonstrated SGH10 to be hyper-mucoid and highly resistant to human serum[7]. Here we confirmed its virulence potential in the murine model via oral infection of C57BL/6 mice, which resulted in translocation to the liver, lungs and spleen 48 h post-inoculation (see Methods and Fig. 3a). We performed additional long and short-read sequencing of the SGH10 genome (see Methods), yielding complete circular sequences for the chromosome (5485,114 bp, 57.43% GC content) (Fig. 3b) and virulence plasmid (pSGH10; 231,583 bp, 50.15% GC) (Fig. 3c). The annotated genome sequence was deposited in GenBank under accessions 'CP025080' (chromosome) and 'CP025081' (virulence plasmid), and the strain submitted to the National Collection of Type Cultures in the United Kingdom (NCTC number 14052).

## Discussion

The earlier comparative genomic studies of up to 27 CG23[15,16] included only four genomes outside of CG23-I, which somewhat obscured the diversity and population structure of the clonal group, and provided insufficient temporal signal to reconstruct evolutionary dynamics. Here, the expanded genome collection revealed that the majority of human clinical isolates of CG23 belong to a clonally expanded sublineage, CG23-I, which emerged in the early 20th century (Fig. 1). This predates the identification of serotype K1 Kp as a significant cause of liver abscess by over 50 years[37,38], suggesting it has been circulating undetected for many decades. We also found evidence that the entire population of CG23 associated with human infections has been expanding since its emergence, but that the CG23-I sublineage has undergone particularly accelerated population growth associated with rapid dissemination to five of the six inhabited continents (Supplementary Fig. 3).

Our genomic analyses suggest some potential mechanisms for the success of the CG23-I sublineage in the human population. The colibactin synthesis locus *clb* (also referred to as *pks*), encoded downstream of the *ybt* locus in ICEKp10, is its most notable feature. Colibactin is a hybrid nonribosomal peptide-polyketide that has a genotoxic effect on host cells by cross-linking DNA and inducing double-strand DNA breaks[19,39,40]. It was first discovered in *E. coli*[40] but has since been reported in 3.5–4% of *K. pneumoniae*[14,41], in which it has also been shown to induce DNA double-strand breaks in HeLa cells[41]. Genotoxicity of colibactin in ST23 *Kp* has been demonstrated for strain 1084 (which belongs to CG23-I, but lacks the virulence plasmid-encoded *iro*, *iuc* and *rmpA/rmpA2* loci, see Figs. 1 and 2b), both in vitro in mouse liver cells and in vivo in liver parenchymal cells of orally infected BALB/c mice[19]. In these experiments, the genotoxic effect of the ST23 *Kp* strain was clearly attributed to colibactin production, using isogenic Δ*clbA* and complemented mutants[19]. Subsequent infection experiments using the same strains revealed that loss of colibactin caused a reduction in dissemination to the blood, liver, spleen and brain[42]. Crucially, this work also indicated that the Δ*clbA* mutant was attenuated in its ability to colonise the intestinal mucosa, which is considered a critical prerequisite for invasive disease[42]. Similarly, colibactin has been shown to promote gut colonisation and to be essential for disseminated infection by *E. coli*[40]. Therefore, we speculate that acquisition of the *clb* locus promotes gut colonisation and/or mucosal invasion of CG23-I *Kp*, leading to enhanced transmissibility and increased virulence relative to ICEKp + clb- strains, which may explain the dissemination of the lineage and its dominance amongst hypervirulent infections globally. Infection experiments using the ICEKp + clb- strain NTUH-K2044 show that it is highly virulent despite the lack of colibactin[13], but virulence in this strain is likely also influenced by additional

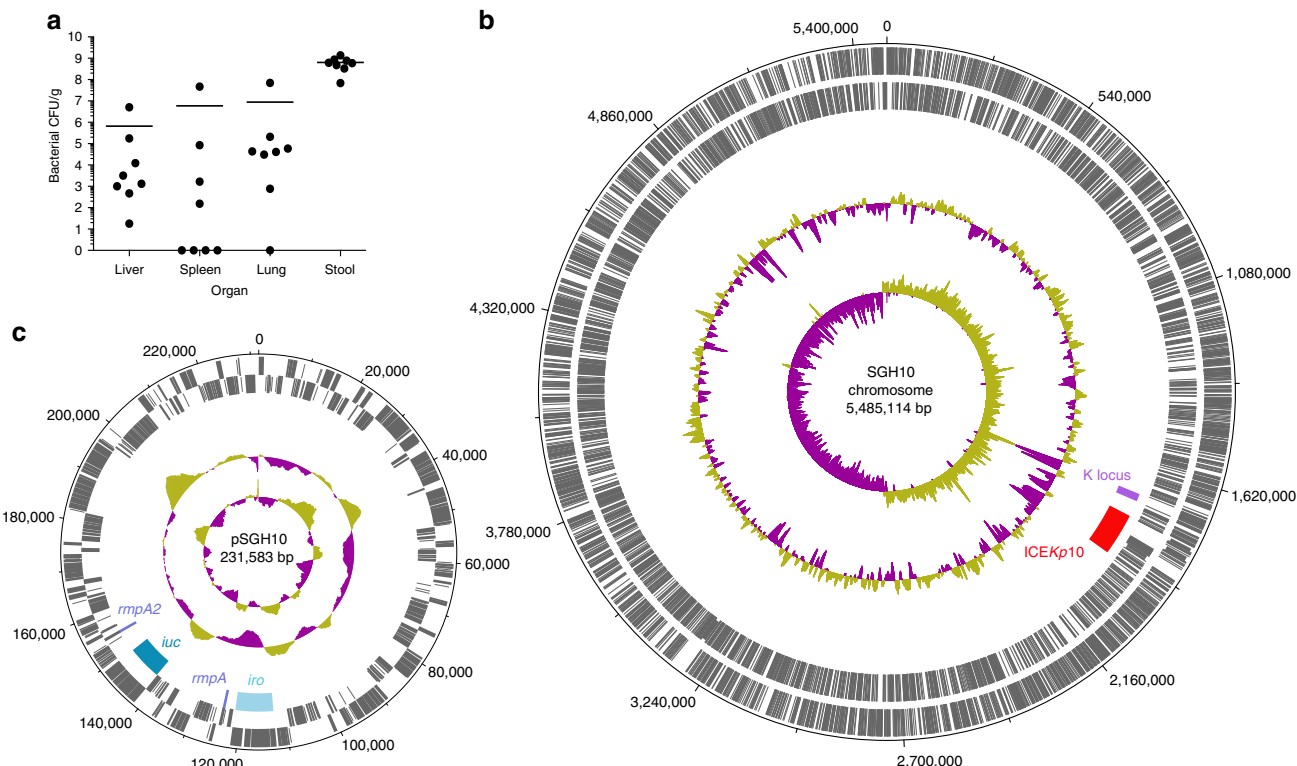

**Fig. 3** Virulence and genomic properties of proposed CG23-I strain SGH10. **a** Bacterial burden (CFU/g) in organs and stool of n = 8 mice 48 h following oral infection, bars indicate mean values. **b** SGH10 chromosome and **c** plasmid. Tracks shown are (from inner to outer): GC skew (G − C/G + C), G+C content, key capsule and virulence features as labelled, coding sequences on reverse strand and coding sequences on forward strand

copies of the *iro* and *rmpA* virulence loci that are not found in the majority of CG23 *Kp* (Figs. 1 and 2, also see below for discussion of other virulence determinants). Concerningly, colibactin also increases the likelihood of serious complications, such as metastatic spread to the brain[42] and potentially tumorigenesis[43,44]. Colibactin-positive *Kp* is particularly common in Taiwan, where it has been reported in 17–25% of non-abscess infections and significantly associated with K1 strains (which are most likely ST23)[43,45]. In this setting pyogenic liver abscess, particularly caused by *K. pneumoniae*, has been shown to be a significant risk factor for colorectal cancer[43,46] and liver cancer[47]. K1 and/or ST23 *Kp* gut colonisation rates up to 10% have been reported across East Asia[48,49]. The prevalence of colibactin among these strains has not yet been investigated in most of these countries, however *Kp* liver abscess has been associated with colorectal cancer across East Asia[50], and based on our genome data we predict the majority of the strains in question will belong to the predominant *clb* + CG23-I sublineage.

The only other gene content changes characteristic of CG23-I were loss-of-function mutations in the fimbrial protein KpcC (via nonsense mutation) and the ethanolamine transporter Eat (via IS insertion). Whether these differences in surface structures or metabolic pathways reflect adaptive selection, or simply loss-of-functions that are not advantageous to the clone, remains a topic for future research. *Kp* has two operons involved in ethanolamine metabolism: *eutRKLCBAHGJENMDTQP* and *eat-eutBC*. The former is typically associated with facultative anaerobes that live in the gut and/or mouth, and the latter with obligate aerobes that may be found in the environment;[51] hence the loss of Eat may reflect adaptation to the host-associated, anaerobic niche. A recent metabolic study of *Kp* growth under aerobic conditions included two ST23 strains NTUH-K2044 (intact *eat*) and SB4385 (CG23-I, disrupted *eat*). Both grew on ethanolamine with no apparent differences in efficiency[52], indicating no detrimental effect of the loss of Eat in CG23-I strains during aerobic growth, hence the available evidence indicates this change is unlikely to be of particular functional significance to the lineage.

The K1 capsule synthesis locus KL1, microcin ICE and virulence plasmid encoding aerobactin, salmochelin and *rmpA* (Fig. 1, Supplementary Fig. 6) were present in most strains, suggesting they were in the ancestor of CG23 and have been selectively maintained. In contrast, distinct variants of ICE*Kp* carrying *ybt* have been acquired on multiple separate occasions after divergence from the CG23 ancestor. Notably only one such acquisition —that of *clb*-positive ICE*Kp10* in CG23-I—was associated with detectable clonal expansion, suggesting positive selection for colibactin production (discussed above). However, given that current genome collections are biased towards clinical strains, we must acknowledge the possibility that CG23 representing other sublineages with or without yersiniabactin and/or colibactin are circulating more frequently among asymptomatic carriers. Indeed, both asymptomatic carriage isolates in our study lacked ICE*Kp*. Asymptomatic gastrointestinal carriage of K1 CG23 has also been reported previously, with up to 10% prevalence in Asian populations[48,49], but whether these strains carry ICE*Kp10* or belong to CG23-I is unknown. On the other hand, three liver abscess strains in our collection also lacked ICE*Kp*, and it was recently reported that the 1932 Murray Collection isolate lacking ICE*Kp* is highly virulent in the *Galleria mellonella* infection model[23]. Hence although *ybt* has been implicated as a key determinant of virulence in mouse models[11,12] and has been positively associated with invasive infections in population studies[9,14], possession of *ybt* does not appear to be a strict requirement for invasion. In addition, CG23-I strain 1084, which lacks the virulence plasmid but carries *ybt* and *clb* in ICE*Kp10* (Figs. 1 and 2b), was isolated from human liver abscess and is

reportedly highly invasive in murine models of pneumonia and liver abscess[19,53]. Taken together these data support the notion that no particular virulence factor of CG23 is necessary or sufficient for invasive disease. Future testing using the *G. mellonella* and/or murine models may shed further light on the interactions between these virulence determinants.

The emergence of AMR CG23 is a significant potential health threat. Treatment of pyogenic liver abscess relies on drainage of the abscess and effective antimicrobial therapy[1]. AMR has been occasionally reported in human isolates of CG23[20–22], and our genomic analysis confirmed that acquisition of AMR plasmids was rare, with distinct plasmids identified in just three sporadic strains from human infections[15,16] and no evidence of long-term plasmid maintenance or transmission amongst human isolates (Fig. 1). This is in contrast to the frequency, diversity and apparent stability of AMR plasmids found in other *Kp* clones such as CG258 or CG15[34,54–56], which also show evidence of frequent ICE*Kp* integration[14]. Overall, our population genomics data suggest there may be barriers to plasmid acquisition and maintenance in CG23 that have so far protected us against the emergence of widespread AMR in this hypervirulent clone. These may include the hyper-expression of K1 capsule, which may provide a physical barrier against transformation and conjugation and also CRISPR/Cas systems, which defend against foreign DNA that does manage to penetrate the cell. The significant association between upregulated capsule production and reduced uptake of DNA has previously been documented in *Streptococcus pneumoniae*[57]. Notably, these mechanisms could also explain the lack of homologous recombination in CG23, which is common in other clonal groups[35]. However, the occasional acquisition of plasmids, including AMR plasmids, in CG23 shows that these barriers are not complete and the maintenance of an IncFII AMR plasmid in the equine clade over 20 years (Supplementary Figs. 6 and 8) shows that long-term stability of AMR plasmids is possible in CG23. Hence our data indicate we must anticipate and carefully monitor for the emergence of stable AMR in CG23.

Taken together our findings have important implications for future study of hypervirulent *Kp*. Firstly, we suggest that epidemiological studies of *Kp* liver abscess infections should investigate and report not only chromosomal MLST and virulence plasmid marker genes such as *rmpA*, but also the *ybt* and *clb* loci, which can be used as markers for the CG23-I sublineage. These loci can be rapidly identified in genome data using our *Klebsiella* genotyping tool, Kleborate (https://github.com/katholt/Kleborate) or can be identified by conventional PCR (primers in Lee et al.[7]). Secondly, our observations regarding the population structure and virulence loci show that the two strains that have been used as reference genomes and/or experimental models for hypervirulent ST23 associated with liver abscess are rather atypical (Fig. 1): NTUH-K2044[17,19] does not belong to the dominant sublineage CG23-I and lacks colibactin, while 1084[19,30] lacks the virulence plasmid. We therefore suggest that future studies aiming to explore the virulence, transmissibility or evolution of CG23 *Kp* should consider genome-based profiling prior to the selection of clinical isolates for experimental work. In addition, we present the human liver abscess isolate SGH10 as an open access reference strain for future studies into hypervirulent CG23 *Kp* associated with liver abscess (strain available under NCTC 14052; complete genome available under GenBank accessions 'CP025080', 'CP025081'). SGH10 exemplifies the most clinically relevant CG23 genotype i.e. that of the globally expanded CG23-I sublineage, including intact copies of all virulence loci. SGH10 is hypermucoid and serum-resistant, is capable of causing disseminated infection in a murine model and in the otherwise healthy human patient from which it was isolated. Finally, we

advocate that researchers, clinicians and public health authorities co-ordinate efforts to improve the surveillance of hypervirulent *Kp* and facilitate rapid identification of emerging threats, such as convergence of hypervirulence and AMR, that may otherwise remain undetected for decades.

## Methods

**Genome collection**. All CG23 genomes that were available at the time of study were included in this work. We identified 83 CG23 genome sequences from our curated collection (six finished and 77 draft genomes, see Supplementary Data 1) [14,26]. These included 43 genomes from four of our own previous studies and 40 publicly available genomes from seven published studies, Genbank and the NCTC3000 genomes project (as described previously [14,26] and including the 27 genomes analysed by Struve et al.) [16]. We also sequenced 15 novel genomes that were identified as ST23 by MLST (02A029, 12A041 and 16A151 [3]) or identified as K1 by classical serotyping and then confirmed as ST23 by genome sequencing (12 strains isolated between 1980 and 1987 from samples from horses; see Supplementary Data 1). DNA was extracted by the phenol–chloroform method, libraries prepared using Nextera technology and paired end reads of either 100 bp (Illumina HiSeq 2000) or 300 bp (Illumina MiSeq) were generated. In total 83 of the 98 genomes represented isolates from human infections (38 liver abscess; 16 bacteraemia with/without liver abscess; 5 pneumonia; 15 miscellaneous; 7 unknown) or asymptomatic carriage (n = 2), collected from Africa, Asia, Australia, Europe and North America between 1932 and 2015 (see Supplementary Data 1). One genome was from an unknown source. The remaining genomes represented *Kp* isolated from horses in France between 1980 and 2004 that were originally selected for whole-genome sequencing solely on the basis of K1 capsule type and were subsequently found to belong to CG23. Accession numbers for all 98 genomes included in this work are listed in Supplementary Data 1 (note though that the hybrid strain CAS686 was excluded from most analyses, as explained below).

**Characterisation of novel CG23-I reference strain SGH10**. We previously sequenced SGH10 (liver abscess isolate, Singapore 2015) via Illumina MiSeq [7]. Here we generated additional short read (Illumina MiniSeq) and long read (Oxford Nanopore) sequences (see Supplementary Methods) and constructed a high quality finished genome sequence using our hybrid assembler Unicycler [58]. Virulence of SGH10 was assessed using a murine oral infection model as previously described [7] with slight modification (see Supplementary Methods). Animal studies were ethically reviewed and approved by the National University of Singapore (NUS) Institutional Animal Care and Use Committee, with the approval number L2015–0135, according to the National Advisory Committee for Laboratory Animal Research (NACLAR) Guidelines.

**Genotyping, yersiniabactin ICE*Kp* and K-locus typing**. Where available (n = 89), short-read sequence data were de novo assembled using Unicycler v0.3.0b [58] with SPAdes v3.8.1 [59] and annotated with Prokka v1.11 [60]. SRST2 [61] was used to (i) determine chromosomal, yersiniabactin and colibactin multi-locus sequence types (STs) [6,14] and (ii) screen for virulence genes included in the BIGSdb [15], acquired AMR genes in the ARG-Annot database [62] and plasmid replicon genes in the PlasmidFinder database [63]. Nine of the publicly available genomes were available only as pre-assembled sequences. For these we used Kleborate (https://github.com/katholt/Kleborate) to determine MLST, virulence and AMR information, and identified plasmid replicons separately by BLASTn search of the assemblies (coverage ≥ 90%, identity ≥ 90%; databases as above). Genomes with novel alleles were submitted to the *Kp* BIGSdb (http://bigsdb.pasteur.fr/klebsiella/klebsiella.html) for assignment of novel allele and ST numbers.

Yersiniabactin ICE*Kp* structures were predicted for all genome assemblies using Kleborate [14] and confirmed by manual inspection using the Artemis genome browser [64]. ICE*Kp* integration sites were determined (from the four possible chromosomal tRNA-Asn sites) using the Bandage assembly graph viewer [65] and BLASTn searches. Microcin E492 ICE sequences were extracted based on their flanking direct repeats and BLASTn searches were used to confirm their integration at tRNA-Asn site 2. K-loci were typed from the assemblies using Kaptive [26].

**Phylogenetic inference and molecular dating analyses**. Sequence reads were mapped to the NTUH-K2044 reference chromosome (accession: 'AP006725') and single nucleotide variants were called using the RedDog v10b pipeline (https://github.com/katholt/reddog) as described in Supplementary Methods. The previous study by Struve et al. [16] showed that isolate CAS686 (ST260) is a hybrid recombinant strain, for which only half of the genome is closely related to CG23. Our mapping data confirmed this, and we further compared the CAS686 genome to our non-CG23 *Kp* collection [14,26] to identify the potential donor lineage (see Results and Supplementary Fig. 4 for further details.) We therefore excluded CAS686 from our main phylogenetic and comparative analyses of CG23, which focus on the remaining 97 CG23 genomes, amongst which 6838 variant sites were identified. The alignment of these sites was screened for further recombination using Gubbins [66], which did not detect any plausible recombination events. We used the complete 6838 site alignment to infer a ML phylogeny using RAxML v8.1.23 [67] with

the GTR+ nucleotide substitution model. The reported ML tree is that with the highest likelihood out of five independent runs. To assess branch support, we conducted 100 non-parametric bootstrap replicates using RAxML. The ML tree and alignment were used as input to FastML [68] for ancestral state reconstruction along the tree.

We used TempEst [69] to investigate the relationship between root-to-tip distances in the ML tree and year of isolation. We then used two different methods to infer the evolutionary rate and estimate time to MRCA (see Supplementary Methods for details). The first method consists of least-squares dating, implemented in LSD v0.3 [70], using as input the ML tree and year of isolation data. We then analysed the data using the Bayesian framework in BEAST v1.8 [71], using two clock models (strict and uncorrelated log normal (UCLD)) and two demographic models (constant population size and Bayesian skyline), as detailed in Supplementary Methods. UCLD clock with constant population size was selected as the best model. We additionally used BEAST to test whether population growth rates differed among lineages (see Supplementary Methods).

**Structural rearrangements, pan-genome and CRISPR analyses**. The chromosomal organisation of the six publicly available completely assembled ST23 genomes (NTUH-K2044, 1084, NCTC9494, ED2, ED23 and RJF999; accessions in Supplementary Data 1) and our newly completed SGH10 genome were subjected to multiple alignment and visualisation using Mauve [72]. All 97 genome assemblies were screened for the presence of phage using PHAST [31]. The pan-genome was investigated using Roary [73] with the Prokka-annotated genomes as the input and ≥95% amino acid identity for protein clustering. A single representative sequence for each gene cluster was compared to the putative phage sequences and the novel pSB4816 sequence (see Supplementary Methods) using BLASTn, in order to identify the components of the pan-genome belonging to phage and pSB4816, respectively. CRISPR arrays were identified using the CRISPR Recognition Tool [74] and *cas* operon genes were identified by tBLASTx (id ≥ 90%, coverage ≥ 28% compared to those annotated in the NTUH-K2044 genome). CRISPR spacer sequences were extracted and clustered at 100% nucleotide identity using CD-HIT-EST [75].

**Plasmid analyses**. To assess the conservation of the pK2044 virulence plasmid, sequence reads were mapped to the pK2044 reference plasmid (accession: 'AP006726.1') using RedDog, and each annotated gene was counted as present in a given isolate if the reads covered ≥ 90% of the gene length at read depth ≥ 5. Putative AMR plasmids were investigated using the Bandage assembly graph viewer [65] as detailed in Supplementary Methods. To generate a reference sequence for the plasmid in the equine clade, we extracted from the assembly graph of isolate SB4816 a putative novel plasmid sequence comprising three contigs that carried all the acquired AMR genes, was separated from the chromosome assembly graph, and displayed 4-fold read depth compared to the chromosomal sequences. We generated long reads for SB4816 as described for SGH10, yielding sufficient reads to confirm the orientation of the three contigs and resolve a complete circular plasmid sequence, pSB4816. Conservation of this plasmid amongst the other equine isolates was assessed by mapping each read set to the pSB4816 sequence as above. Unfortunately we were not able to extract the complete sequences of the three AMR plasmids from public genome data due to complexities in the assembly graphs, but we were able to identify the putative plasmid replicon types and associated plasmid STs (see Supplementary Methods).

**Data availability**. Accession numbers for all genome data included in this work are summarised in Supplementary Data 1. Illumina sequence reads for the 15 newly sequenced equine isolates have been deposited in the NCBI sequence read archive under accession 'PRJNA391004'. The MDR plasmid sequence pSB4816 was deposited in GenBank (accession: 'MF363048'). The finished genome sequence for SGH10 (BioSample; 'SAMN06112188') was deposited in GenBank (accessions 'CP025080.1', 'CP025081') and the strain was deposited in the UK National Collection of Type Cultures (NCTC 14052); novel Illumina and Oxford Nanopore sequence reads for this isolate are available in the sequence read archive under accessions 'SRR6266394', 'SRR6266393', 'SRR6266392' and 'SRR6307304'. The Bayesian phylogenetic tree, annotated with strain information, is available for download and interactive visualisation at https://microreact.org/project/r1_dSC9w-.

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

## Acknowledgements

We thank Jonathan Cotrupi for microbiological analysis of horse strains from France, Ulises Garza-Ramos and Alexis Criscuolo for help in genomic analyses, and Carla Rodrigues for assistance with Nanopore sequencing of strain SB4816. We also thank the team of the curators of the Institut Pasteur MLST system (Paris, France) for importing novel alleles, profiles and/or isolates at http://bigsdb.pasteur.fr. This work was supported by the NHMRC of Australia (fellowship #1061409 to KEH), National University of Singapore (grant number NUHSRO/2014/068/AF-New Idea/03 to Y.H.G.), and the Bill and Melinda Gates Foundation, Seattle.

## Author contributions

M.M.C.L. and K.L.W. analysed data and wrote the paper. S.D., R.R.W., S.A. analysed data. L.M.J., V.P., Y.H.G. and C.H.H. performed lab experiments. Y.H.G., S.A., J.S.M., S. K., T.H.K. and S.B. contributed isolates for sequencing. S.B. and K.E.H. conceived the study, analysed and interpreted data and contributed to manuscript writing.

## Additional information

**Competing interests:** The authors declare no competing interests.

