## [Peer Review File · Nature Communications]

Reviewer #1 (Remarks to the Author):

In this manuscript, the authors undertake a broad genomic investigation of a *Klebsiella pneumoniae* (Kp) clonal group, CG23, that is of significant public health importance. This work has two distinct primary scopes: (1) to delineate the evolutionary timeline of CG23 and (2) to demarcate a sublineage of this group, CG23-I, that is overrepresented as an etiological agent of liver abscess. The authors also propose a new strain, SGH10, as a model CG23-I strain for future study and establish its ability to cause a productive infection in a murine model of oral Kp infection. A strength of this manuscript is the authors' exhaustive genomic analysis in conjunction with the inclusion of a large number of CG-23 genomes. Additionally, the manuscript is well written and organized, and clearly prepared with care. The manuscript could be improved through comparison of their proposed CG23-I reference strain with other frequently used CG23 strains in their animal studies and by adding clarifying language to genomic analyses to increase accessibility to a broad audience interested in Kp pathogenesis. Overall, the manuscript provides novel insight into the the evolutionary history of CG23 and greater detail on the predominate lineages of hypervirulent Kp that are of high public health concern and scientific interest.

Specific concerns:

Major concerns -

1. Lack of comparison of virulence of their proposed reference strain to other well-characterized CG23 isolates. The observation that the CG23-I group is the most prevalent in their strain collection is important in considering what strains to use in models of hypervirulent Kp infection. The authors propose that the presence of colibactin in this group is critical to its success as a pathogen. However, they also note that no virulence gene appears to be necessary or sufficient to cause invasive disease, and the importance of colibactin in a murine model was seen in a strain that lacked other well characterized virulence factors on the pK2044 plasmid. The rationale to use SGH10 as a model strain instead of NTUH-K2044 and 1084 would be strengthened by showing equivalent or greater virulence. Various murine models of gastric colonization and dissemination have already been established for CG23 and non-CG23 Kp strains (Caballero et al., 2015; Hsieh et al., 2008; Panpetch et al., 2018). The data shown for SGH10 is bacterial enumeration at peripheral tissues and stool following oral infection. In the work of Hsieh et al NTUH has an LD50 of 10e5 by gastric inoculation, and an attenuated tonB mutant has an LD50 of ~1e6. The significantly higher dose of 1e8 for SGH10 suggests it may be significantly less virulent, or at least its relative virulence is unclear.

2. Need for additional clarity for interpretation of genomic data. This manuscript provides novel and exciting insight into the genetic lineage of hypervirulence CG23 Kp, especially estimating its date of emergence and delineating important differences in gene content. However, the description of results and methods is written with little explanation for non-experts in evolutionary genomics. In order to communicate to a broad audience, the explanations of Figures 1, 2, S2, and S3 could be stated more plainly and the more technical details left in supplemental results and methods. Additionally, the authors emphasize the "...rapid global dissemination..." of CG23-I Kp strains in the title and introduction, yet this point is not addressed in the manuscript other than the fact that the representative Kp strains were isolated from different continents. Either more information needs to be extrapolated from the genomic data and discussed, or this section of the title needs to be changed.

Minor concerns/questions -

1. Figure S6A - given that there are multiple virulence factors on pK2044, it would valuable to graphically highlight the other major virulence determinants (e.g. heavy metal resistance genes, adhesins, membrane-bound iron transport systems) located on this plasmid.
2. A significant portion of the data is focused on equine Kp, which the authors identify as a clade of the CG23-I sublineage, rather than a divergent lineage. What are the relevance of these findings

to human Kp ecology/pathology? Additional discussion of this relationship would be valuable given the presence of antibiotic resistance genes in these strains.

3. Figure S2: Noting NTUH and other strains of interest would help the reader compare this ML tree and the bayesian tree in figure 1.
4. Figures 2A-C are called out but not interpreted in the text. It may be more appropriate to move to supplemental material.
5. Line 99: This discussion of a hybrid isolate is an awkward beginning to the results and takes away from the main objectives of the study.
6. It is not clear how extinction of lineages is defined in a sample of 100 isolates. It seems possible that some lineages persist in nature and were not sampled.
7. The experimental evidence of colibactin importance is from a strain without the virulence plasmid. This should be discussed in the context of NTUH, which is highly virulent in the same animal model but without colibactin. Lines 308-317 could be moved earlier to the discussion of colibactin.

Reviewer #2 (Remarks to the Author):

This manuscript by Lam et al. describes an updated evolutionary history of hypervirulent *K. pneumoniae* CG 23. The study takes advantage of 83 previously published genomes and an additional newly sequenced 15 genomes. Bayesian analyses suggested that the origin of the CG23 lineage dates back to the 1870s and that the lineage encompasses two major clades. This suggests that the lineage circulated prior to the onset of relatively widespread liver abscesses. While several sublineages were detected within CG23, clade CG23-I was found to dominate, potentially due to recent accelerated expansion. This was attributed to these isolates harboring and ICE, encoding for colibactin. It has been suggested that colibactin promotes gut colonization and potentially also dissemination of organisms to other organs. Overall, this is a nicely executed study that provides novel insights into the evolutionary history of hypervirulent Kp and CG23; however given the limited addition of novel sequences in parts it recapitulates older findings and appears somewhat limited in scope.

Specific comments:

1. One concern with this analysis is a potential bias in the sample selection and the sequenced sample not representing the full population structure of CG23. How were the 15 sequenced isolates selected? Horse isolates appear to be from an outbreak but others represent a more diverse temporal and geographic collection. Although mainly previously published please describe your isolate / genome collection in more detail in the methods.
2. Given the importance of horizontal spread of AMR, the composition of the plasmids that putatively harbor resistance genes should be resolved by long-range sequencing.
3. Several sections of the discussion section are highly speculative. For example, line 312 – the putative impact of lack of ICEKp. Validation of the proposed functional effect of this locus in the *Galleria* model would significantly strengthen this manuscript.

Reviewer #3 (Remarks to the Author):

This manuscript describes the population genomic of *K. pneumoniae* clonal group 23. Overall the paper is very nicely presented and well written.

Comments

1. Line 47-8; I did not find sufficient data in this MS to support that statement.
2. Line 62-4; it will be nice for readers not familiar with CG23 to include all the STs within this

clonal group.

3. Line 116; To which STs did the 81 isolates belong to?

4. Lines 234-42; without long read sequencing (that was done for the virulence plasmid), it is very difficult to judge the validity of AMR plasmid results. The authors should include a sentence stating that.

Population genomics of hypervirulent *Klebsiella pneumoniae* clonal-group 23 reveals early emergence and rapid global dissemination

Point-by-point response to reviewer comments

Reviewer #1

In this manuscript, the authors undertake a broad genomic investigation of a *Klebsiella pneumoniae* (Kp) clonal group, CG23, that is of significant public health importance. This work has two distinct primary scopes: (1) to delineate the evolutionary timeline of CG23 and (2) to demarcate a sublineage of this group, CG23-I, that is overrepresented as an etiological agent of liver abscess. The authors also propose a new strain, SGH10, as a model CG23-I strain for future study and establish its ability to cause a productive infection in a murine model of oral Kp infection. A strength of this manuscript is the authors' exhaustive genomic analysis in conjunction with the inclusion of a large number of CG-23 genomes. Additionally, the manuscript is well written and organized, and clearly prepared with care. The manuscript could be improved through comparison of their proposed CG23-I reference strain with other frequently used CG23 strains in their animal studies and by adding clarifying language to genomic analyses to increase accessibility to a broad audience interested in Kp pathogenesis. Overall, the manuscript provides novel insight into the evolutionary history of CG23 and greater detail on the predominate lineages of hypervirulent Kp that are of high public health concern and scientific interest.

Major concerns -

Q1. Lack of comparison of virulence of their proposed reference strain to other well-characterized CG23 isolates. The observation that the CG23-I group is the most prevalent in their strain collection is important in considering what strains to use in models of hypervirulent Kp infection. The authors propose that the presence of colibactin in this group is critical to its success as a pathogen. However, they also note that no virulence gene appears to be necessary or sufficient to cause invasive disease, and the importance of colibactin in a murine model was seen in a strain that lacked other well characterized virulence factors on the pK2044 plasmid. The rationale to use SGH10 as a model strain instead of NTUH-K2044 and 1084 would be strengthened by showing equivalent or greater virulence. Various murine models of gastric colonization and dissemination have already been established for CG23 and non-CG23 Kp strains (Caballero et al., 2015; Hsieh et al., 2008; Panpetch et al., 2018). The data shown for SGH10 is bacterial enumeration at peripheral tissues and stool following oral infection. In the work of Hsieh et al NTUH has an LD50 of 10e5 by gastric inoculation, and an attenuated tonB mutant has an LD50 of ~1e6. The significantly higher dose of 1e8 for SGH10 suggests it may be significantly less virulent, or at least its relative virulence is unclear.

Response: Firstly, the rationale to propose SGH10 as a reference for future experimental work on CG23 virulence, rather than NTUH-K2044 or 1084, is based on our novel findings from the genomic analyses which clearly show that the latter two strains are atypical in terms of their allelic variation across the genome (position in the phylogeny) and the virulence determinants that they possess (NTUH-K2044 lacking the typical colibactin ICE, 1084 lacking the virulence plasmid). In contrast, SGH10 is genetically representative of the dominant CG23-I lineage, which our data show cause the majority of CG23-associated liver-abscess disease. Thus SGH10 is a more clinically relevant model strain, irrespective of its relative virulence potential compared to other non-representative strains.

Secondly, whilst we agree that it could be interesting to directly compare the virulence of these strains in a single set of experiments, we consider this to be far beyond the scope of the current manuscript, which concerns the evolution and genetic diversity of CG23; and note that we do not have the strains NTUH-K2044 and 1084 which are not publicly available.

Additionally, the reviewer correctly points out differences between the NTUH-K2044 LD50 reported in the literature and that of SGH10 in our experimental system; however this is because the published studies of NTUH-K2044 (Hsieh et al 2008), and indeed other studies using non-ST23 isolate CG43, used BALB/c mice or derivatives thereof, whereas we used C57BL/6 mice (which we chose to work with due to the availability of knockout mutant mouse strains for subsequent work). BALB/c mice are known to be more susceptible to certain infections caused by several bacterial pathogens, due to a Th2 bias in BALB/c versus a Th1 bias in C57BL/6 (see Sellers et al 2012 *Veterinary Pathology* 49(1) 32-43), hence a lower LD50 would be expected even for the same inoculating strain. Additionally, there is evidence that these mouse strains carry distinct gut microbiota (Jiang et al *AMB Express* 2018, 8:31), which may further influence infective or lethal dose. Hence the difference in LD50 in these two mouse strains cannot be interpreted as demonstrating a difference in virulence potential of the bacterial strains. We also note that SGH10 was isolated from a 35 year old male with no diabetes or any underlying disease and yet was able to disseminate to the liver. SGH10 therefore represents a highly virulent strain in humans.

These additional details have been added to the Results text to clarify this point:

“We therefore propose strain SGH10 as a reference strain for experimental and genomic studies of hypervirulent CG23 *K. pneumoniae* associated with human liver abscess because it belongs to the predominant CG23-I sublineage responsible for the majority of CG23-associated liver abscess infections, and it has all of the common virulence loci intact with no atypical accessory genes (see Fig. 1 and Supplementary Fig. 9). SGH10 was isolated from a 35-year old liver abscess patient in Singapore in 2014 with no diabetes and no detectable underlying disease, and we have previously demonstrated it to be hypermucoid and highly resistant to human serum.”

We have also amended the text in the final paragraph of the Discussion to the following:

“In addition, we present the human liver abscess isolate SGH10 as an open access reference strain for future studies into hypervirulent CG23 *Kp* associated with liver abscess (strain available under NCTC 14052; complete genome available under GenBank accessions CP025080, CP025081). SGH10 exemplifies the most clinically relevant CG23 genotype i.e. that of the globally expanded CG23-I sublineage, including intact copies of all virulence loci. SGH10 is hypermucoid and serum-resistant, and is capable of causing disseminated infection in a murine model and in the otherwise healthy human patient from which it was isolated.”

Q2. Need for additional clarity for interpretation of genomic data. This manuscript provides novel and exciting insight into the genetic lineage of hypervirulence CG23 *Kp*, especially estimating its date of emergence and delineating important differences in gene content. However, the description of results and methods is written with little explanation for non-experts in evolutionary genomics. In order to communicate to a broad audience, the explanations of Figures 1, 2, S2, and S3 could be stated more plainly and the more technical details left in supplementary methods. Additionally, the authors emphasize the "...rapid global dissemination..." of CG23-I *Kp* strains in the title and introduction, yet this point is not addressed in the manuscript other than the fact that the representative *Kp* strains were isolated from different continents. Either more information needs to be extrapolated from the genomic data and discussed, or this section of the title needs to be changed.

Response: We thank the reviewer for raising this important point, and have made a number of minor changes throughout these sections (examples listed below), moving much of the details to the supplementary section and scaling back the evolutionary genomics jargon in the main text, particularly that pertaining to what was previously Fig 2 (now moved to Supplementary Fig 2) summarising the dating methods and results:

“We detected a strong temporal signal in the CG23 genome alignment (see **Supplementary Fig. 2A-C, Supplementary Methods**), sufficient to estimate

evolutionary rates and dates for the most recent common ancestors (MRCAs) of key CG23 lineages with BEAST (best-fitting Bayesian model was the UCLD constant model, see **Supplementary Methods**). The mean evolutionary rate for CG23 was estimated to be 3.40×10^{-7} substitutions site⁻¹ year⁻¹ (95% HPD; 2.43×10^{-7} - 4.38×10^{-7}). The MRCAs for the entire CG23 population, CG23-I and equine sublineage nested within CG23-I were estimated to be 1878 (95% HPD; 1827-1915), 1928 (95% HPD; 1908-1953) and 1972 (95% HPD, 1961-1975) respectively. We hypothesised that the successful spread of CG23-I in the human population may have been associated with a population expansion; i.e. an increase in diversification rate and effective population size, compared to the rest of the CG23 population. We used an epidemiological model (birth-death) to compare these dynamics between the different lineages (see **Supplementary Methods**), which demonstrated population decline in the horse lineage and expansion of the entire human-associated CG23 population (**Supplementary Fig. 3A, C**), particularly among CG23-I, which was associated with five times the rate of expansion than the rest of the population (**Supplementary Fig. 3B, D**). Importantly, there was significant overlap in the Bayesian parameter estimate distributions from four alternative models, and with estimates derived from an alternative method known as least-squares LSD analysis (**Supplementary Fig. 2D-F**, see **Supplementary Methods**). This provides confidence that CG23 emerged some time in the late 19th century and the globally distributed CG23-I emerged in the 1920s.”

Other examples:

“Comparative analysis of the 97 CG23 genomes (**Supplementary Table 1**) identified no recombination events (besides the known hybrid strain CAS686²⁸ which was excluded from analysis, see below), and showed low nucleotide divergence across core chromosomal genes, with a median pairwise distance of 233 single nucleotide polymorphisms (SNPs; range 1-444 SNPs), and 0.0045% nucleotide divergence (range 0.000019% - 0.0086%).”

“Phylogenetic analyses indicated that CG23 has a number of sub-lineages separated by deep-branches, one of which (labelled CG23 sublineage I, CG23-I) has become globally distributed (**Fig. 1, Supplementary Fig. 1**). The CG23-I clade was strongly supported by two independent analysis methods (>99% posterior support in Bayesian tree, 100% bootstrap support in ML tree). This clade comprised 81 isolates (83.5% of all CG23) collected from Asia, Australia, North America, Europe and Africa (**Fig. 1**), including 82% of all liver abscess strains (**Supplementary Table 1**). Neither of the two oldest strains in our analysis, M109 (1932, Murray Collection³⁵) and NCTC9494 (1954, human sputum), were part of CG23-I (**Fig. 1**); nor was NTUH-K2044, the first sequenced ST23 strain that has served as a reference for much of the reported experimental and genomic work on ST23²⁸⁻³⁰.”

“We used TempEst⁹³ to investigate the relationship between root-to-tip distances in the ML tree and year of isolation. We then used two different methods to infer the evolutionary rate and estimate time to MRCA (see **Supplementary Methods** for details). The first method consists of least-squares dating, implemented in LSD v0.3⁹⁴, using as input the ML tree and year of isolation data. We then analysed the data using the Bayesian framework in BEAST v1.8⁹⁵, using two clock models (strict and uncorrelated log normal (UCLD)) and two demographic models (constant population size and Bayesian skyline), as detailed in **Supplementary Methods**. UCLD clock with constant population size was selected as the best model. We additionally used BEAST to test whether population growth rates differed among lineages (see **Supplementary Methods**).”

Regarding the "...rapid global dissemination..." of CG23-I Kp strains in the title and introduction, the rapid dissemination refers to the accelerated population growth of CG23-I compared to the rest of CG23 as demonstrated by population dynamics analysis with the birth-death model. This has been made clearer in the main text as follows:

"We used an epidemiological model (birth-death) to compare these dynamics between the different lineages (see **Supplementary Methods**), which demonstrated population decline in the horse lineage and expansion of the entire human-associated CG23 population (**Supplementary Fig. 3A, C**), particularly among CG23-I, which was associated with five times the rate of expansion than the rest of the population (**Supplementary Fig. 3B, D**). Importantly, there was significant overlap in the Bayesian parameter estimate distributions from four alternative models, and with estimates derived from an alternative method known as least-squares LSD analysis (**Supplementary Fig. 2D-F**, see **Supplementary Methods**). This provides confidence that CG23 emerged some time in the late 19th century and the globally distributed CG23-I emerged in the 1920s."

We have also amended the relevant points in the first paragraph of the Discussion:

"We also found evidence that the entire population of CG23 associated with human infections has been expanding since its emergence, but that the CG23-I sublineage has undergone particularly accelerated population growth associated with rapid dissemination to five of the six inhabited continents (**Supplementary Fig. 3**)."

Minor concerns/questions -

Q1. Figure S6A - given that there are multiple virulence factors on pK2044, it would be valuable to graphically highlight the other major virulence determinants (e.g. heavy metal resistance genes, adhesins, membrane-bound iron transport systems) located on this plasmid.

Response: The figure has now been amended to include the positions of other loci encoding resistance to the heavy metals copper, silver and tellurite, and the membrane-bound iron transport system.

Q2. A significant portion of the data is focused on equine Kp, which the authors identify as a clade of the CG23-I sublineage, rather than a divergent lineage. What is the relevance of these findings to human Kp ecology/pathology? Additional discussion of this relationship would be valuable given the presence of antibiotic resistance genes in these strains.

Response: The reviewer raises an interesting question about the relationship between equine *K. pneumoniae* and transmission and infection in humans. We have added some extra text (shown in bold) to highlight the lack of current information in this area:

"*Kp* is considered an important cause of sexually transmitted disease in horses³⁷ but little is known about the molecular epidemiology of this group or sources of infection. The horse isolates included in this study (including genital tract, sperm, foetus and metritis specimens isolated between 1980 and 2004) were originally selected for sequencing only on the basis of K1 serotype, but clustered together within a monophyletic subclade nested within CG23-I (**Fig. 1**), separated from the rest of CG23-I by 83 SNPs (**Supplementary Table 3**). Of note, four non-synonymous mutations arose within genes encoding putative oxidoreductases and two intergenic mutations were within close proximity to predicted oxidoreductase genes.

The nested positioning within CG23-I suggests that CG23 *Kp* may have entered the horse population on a single occasion where it now circulates via sexual contact, and there is no evidence of transmission between humans and horses."

Q3. Figure S2: Noting NTUH and other strains of interest would help the reader compare this ML tree and the bayesian tree in figure 1.

Response: We thank the reviewer for pointing this out and the figure (changed to Supplementary Fig. 1) has now been amended to include labels for the strains with complete genome sequences.

Q4. Figures 2A-C are called out but not interpreted in the text. It may be more appropriate to move to supplemental material.

Response: As per Major Q2 above, we have moved Figure 2 to a supplementary figure (Supplementary Fig. 2).

Q5. Line 99: This discussion of a hybrid isolate is an awkward beginning to the results and takes away from the main objectives of the study.

Response: The introduction to this section of the results has now been re-structured as follows, and the discussion of the hybrid isolate moved down to another section under “*Isolate CAS686 is a hybrid of ST23 and ST281-like Kp*”.

“Comparative analysis of the 97 CG23 genomes (**Supplementary Table 1**) identified no recombination events (besides the known hybrid strain CAS686²⁸ which was excluded from analysis, see below) ...”

Q6. It is not clear how extinction of lineages is defined in a sample of 100 isolates. It seems possible that some lineages persist in nature and were not sampled.

Response: In the context it was used (explaining the interpretation of the birth-death model) ‘extinct’ is the correct technical term, however we recognise this could be confusing/misleading for many readers; hence in line with the response to Q2 we have modified this section and moved the technical details to supplementary methods as suggested.

We have also added a note about the potential of sampling bias towards clinical isolates to influence the detection of sublineages outside of CG23-1. Discussion paragraph 4:

“Notably only one such acquisition – that of *clb*-positive ICE*Kp10* in CG23-I – was associated with detectable clonal expansion, suggesting positive selection for colibactin production. However, given that current genome collections are biased towards clinical strains, we must acknowledge the possibility that CG23 representing other sublineages are circulating more frequently among asymptomatic carriers.”

Q7. The experimental evidence of colibactin importance is from a strain without the virulence plasmid. This should be discussed in the context of NTUH, which is highly virulent in the same animal model but without colibactin. Lines 308-317 could be moved earlier to the discussion of colibactin.

Colibactin is discussed in detail in its own section due to its relevance as a defining feature of CG23-1. As noted in the text, the virulence profile of NTUH-K2044 cannot be considered typical of CG23 as it carries additional siderophores in the ICE*Kp* element which are not present in another other strain. We have added lines to the text to highlight this and to remind readers that the 1084 strain lacks the plasmid-borne virulence loci:

“Genotoxicity of colibactin in ST23 *Kp* has been demonstrated for strain 1084 (which belongs to CG23-I, but lacks the virulence plasmid-encoded *iro*, *iuc* and *rmpA/rmpA2*

loci), both *in vitro* in mouse liver cells and *in vivo* in liver parenchymal cells of orally infected BALB/c mice ³¹.”

...

“Therefore, the acquisition of the *clb* locus likely promotes both gut colonisation and mucosal invasion of CG23-I *Kp*, leading to enhanced transmissibility and increased virulence relative to ICE*Kp*+ *clb*- strains, which may explain the dissemination of the lineage and its dominance amongst hypervirulent infections globally. (Infection experiments using the ICE*Kp*+ *clb*- strain NTUH-K2044 show that it is highly virulent despite the lack of colibactin (Hseih *et al*, J Infect Dis 2008), but virulence in this strain is likely also influenced by additional copies of the *iro* and *rmpA* virulence loci that are not found in the majority of CG23 *Kp* (also see below for discussion of other virulence determinants).)”

Reviewer #2 (Remarks to the Author):

This manuscript by Lam et al. describes an updated evolutionary history of hypervirulent *K. pneumoniae* CG 23. The study takes advantage of 83 previously published genomes and an additional newly sequenced 15 genomes. Bayesian analyses suggested that the origin of the CG23 lineage dates back to the 1870s and that the lineage encompasses two major clades. This suggests that the lineage circulated prior to the onset of relatively widespread liver abscesses. While several sublineages were detected within CG23, clade CG23-I was found to dominate, potentially due to recent accelerated expansion. This was attributed to these isolated harboring and ICE, encoding for colibactin. It has been suggested that colibactin promotes gut colonization and potentially also dissemination of organisms to other organs. Overall, this is a nicely executed study that provides novel insights into the evolutionary history of hypervirulent *Kp* and CG23; however given the limited addition of novel sequences in parts it recapitulates older findings and appears somewhat limited in scope.

Response: The reviewer correctly notes that the majority of individual genomes included in this work are not novel, however combining these data from over a dozen reported studies into a single comparative study of 97 genomes is new. Most importantly, this approach yielded novel findings that have been missed by the previous comparative analyses of smaller genome sets ($n \leq 27$), which did not capture sufficient temporal or genetic variation to elucidate population structure; to allow Bayesian dating analyses; to identify the multiple acquisitions of ICE*Kp*; or to identify the CG23-I sublineage. Furthermore, none of the previous work included any analyses of variation in the CG23 pan-genome, phage content, or CRISPR/Cas systems which are entirely novel aspects of this work. Only by combining all of the available CG23 genomes collected from disparate sources have these analyses been possible. We therefore feel that it is incorrect to imply that the current study is limited in scope simply because it includes a majority of genomes that were previously published.

We have amended the Methods text to highlight that the genomes included here were collated from a diverse range of sources (i.e. that the majority have not been directly compared in previous works):

“We identified 83 CG23 genome sequences from our curated collection (six finished and 77 draft genomes, see Supplementary Table 1 ^{26,38}). These included 43 genomes from four of our own previous studies ^{19,21,27,69} and 40 publicly available genomes from seven published studies, Genbank and the NCTC3000 genomes project (including the 27 genomes analysed by Struve *et al*) ^{28,29,37,68,70-74}.”

Specific comments:

Q1a. One concern with this analysis is a potential bias in the sample selection and the sequenced sample not representing the full population structure of CG23.

Response: All CG23 genomes available at the time of the study were included to facilitate a thorough and in depth investigation into the evolutionary history of the clone – this has now been made clear at the beginning of the methods section.

“All CG23 genomes that were available at the time of study were included in this work. We identified 83 CG23 genome sequences from our curated collection (six finished and 77 draft genomes, see Supplementary Table 1^{26,38}). These included 43 genomes from four of our own previous studies^{19,21,27,69} and 40 publicly available genomes from seven published studies, Genbank and the NCTC3000 genomes project (including the 27 genomes analysed by Struve *et al*)^{28,29,37,68,70–74}.”

However, we recognise that the available genomes are biased towards clinical isolates and agree that this is an important point to highlight in the discussion. The Discussion text has been modified as follows:

“...given that current genome collections are biased towards clinical strains, we must acknowledge the possibility that CG23 representing other sublineages with/without ICE*Kp* +/- *clb* are circulating more frequently among asymptomatic carriers.”

Q1b. How were the 15 sequenced isolates selected? Horse isolates appear to be from an outbreak but others represent a more diverse temporal and geographic collection. Although mainly previously published please describe your isolate / genome collection in more detail in the methods.

Response: The 15 horse isolates were sampled from 1980-2004 and do not represent an outbreak. They were selected for genomic sequencing on the basis of K1 capsule expression only (ie suspicion of belonging to CG23). The following text has been added to the methods section to further describe the genome collection:

“In total 83 of the 98 genomes represented isolates from human infections (38 liver abscess; 16 bacteraemia with/without liver abscess; 5 pneumonia; 15 miscellaneous; 7 unknown) or asymptomatic carriage (n = 2), collected from Africa, Asia, Australia, Europe and North America between 1932 and 2015 (see Supplementary Table 1). One genome was from an unknown source. The remaining genomes represented *Kp* isolated from horses in France between 1980 and 2004 that were originally selected for whole-genome sequencing solely on the basis of K1 capsule type and were subsequently found to belong to CG23.”

The description of the horse subclade in the Results has also been amended to avoid any confusion about these genomes:

“The horse isolates included in this study (including genital tract, sperm, foetus and metritis specimens isolated between 1980 and 2004) were originally selected for sequencing only on the basis of K1 serotype, but clustered together within a single subclade inside CG23-I (Fig. 1).”

Q2. Given the importance of horizontal spread of AMR, the composition of the plasmids that putatively harbor resistance genes should be resolved by long-range sequencing.

Response: We have now generated long reads (via Oxford Nanopore) for our horse isolate SB4816 sufficient to resolve the complete sequence for the equine clade-associated AMR plasmid pSB4816 that is discussed in the detail in the manuscript. The three other AMR plasmids mentioned are public genome data so we have no capacity to resequence them. Note. The Genbank submission for the plasmid is currently being updated and may not yet be available at the time of resubmitting this manuscript.

We have amended the ‘Plasmid analyses’ section of the Methods as follows:

“To generate a reference sequence for the plasmid in the equine clade, we extracted from the assembly graph of isolate SB4816 a putative novel plasmid sequence comprising three contigs that carried all the acquired AMR genes, was separated from the chromosome assembly graph, and displayed 4-fold read depth compared to the chromosomal sequences. We generated long reads for SB4816 as described for SGH10, yielding sufficient reads to confirm the orientation of the three contigs and resolve a complete circular plasmid sequence, pSB4816. Conservation of this plasmid amongst the other equine isolates was assessed by mapping each read set to the pSB4816 sequence as above. Unfortunately we were not able to extract the complete sequences of the three AMR plasmids from public genome data due to complexities in the assembly graphs, but we were able to identify the putative plasmid replicon types and associated plasmid STs (see **Supplementary Methods**).”

Q3. Several sections of the discussion section are highly speculative. For example, line 312 – the putative impact of lack of ICEKp. Validation of the proposed functional effect of this locus in the *Galleria* model would significantly strengthen this manuscript.

Response: In this section of the Discussion we highlight the available evidence regarding the importance of ICEKp containing *ybt* +/- *clb* to virulence, for which there is extensive murine model data and epidemiological data from human infections. Specifically, experiments using murine models clearly implicate *ybt* as an important determinant e.g. Bachman et al. Infect Immun 2011 and Holden et al mBio 2016, and our population based studies have shown a clear association between the presence of *ybt* and invasive infection (Holt et al. PNAS 2015; Lam et al bioRxiv 2017). However, there is also evidence that CG23 *ybt* –ve strains can be highly virulent (here and Wand et al AAC 2015). However, we acknowledge the presentation of this evidence and our conclusions could be written more clearly; and that further systematic knockout studies using the *Galleria* and mouse models may be beneficial although beyond the scope of the current study. The corresponding section of the text has been altered in light of this and other reviewer comments:

“In contrast, distinct variants of ICEKp carrying *ybt* have been acquired on multiple separate occasions after divergence from the CG23 ancestor. Notably only one such acquisition – that of *clb*-positive ICEKp10 in CG23-I – was associated with detectable clonal expansion, suggesting positive selection for colibactin production. However, given that current genome collections are biased towards clinical strains, we must acknowledge the possibility that CG23 representing other sublineages with/without ICEKp +/- *clb* are circulating more frequently among asymptomatic carriers. Indeed, both asymptomatic carriage isolates in our study lacked ICEKp. Asymptomatic gastrointestinal carriage of K1 CG23 has also been reported previously, with up to 10% prevalence in Asian populations^{52,53}, but whether these strains carry ICEKp10 or belong to CG23-I is unknown. On the other hand, three liver abscess strains in our collection also lacked ICEKp, and it was recently reported that the 1932 Murray Collection isolate lacking ICEKp is highly virulent in the *Galleria mellonella* infection model³⁷. Hence although *ybt* has been implicated as a key determinant of virulence in mouse models (Holden mBio 2016; Bachman I&I 2011) and has been positively associated with invasive infections in population studies (Holt PNAS 2015; Lam bioRxiv 2017), possession of *ybt* does not appear to be a strict requirement for invasion. In addition, CG23-I strain 1084, which lacks the virulence plasmid but carries *ybt* and *clb* in ICEKp10, was isolated from human liver abscess and is reportedly highly invasive in murine models of pneumonia and liver abscess^{31,59}. Taken together these data support the notion that no particular virulence factor of CG23 is necessary or sufficient for invasive disease. Future testing using the *G. mellonella* and/or murine models may shed further light on the interactions between these virulence determinants. “

Reviewer #3 (Remarks to the Author):

This manuscript describes the population genomic of *K. pneumoniae* clonal group 23. Overall the paper is very nicely presented and well written.

Comments

Q1. Line 47-8; I did not find sufficient data in this MS to support that statement.

Response: The statement in question (in the abstract) is "... and has the capacity to acquire and maintain AMR plasmids."

The sustained carriage of a plasmid encoding antimicrobial resistance was observed in isolates sampled from horses across a period spanning 1980 to 2004, indicating that at least one sublineage within CG23 (and potentially others) has the ability to maintain such plasmids for decades, as outlined in the Discussion section:

"... the occasional acquisition of plasmids, including AMR plasmids, in CG23 shows that these barriers are not complete and the maintenance of an IncFII AMR plasmid in the equine clade over 20 years (**Supplementary Fig. 6 and 8**) shows that long-term stability of AMR plasmids is possible in CG23. Hence our data indicate we must anticipate and carefully monitor for the emergence of stable AMR in CG23."

Q2. Line 262-4; it will be nice for readers not familiar with CG23 to include all the STs within this clonal group.

Response: The text has been amended to include common sequence types within CG23:

"In particular, CG23 was shown to account for 37-64% isolates in Taiwan¹⁶, Singapore¹⁹ and mainland China^{15,20}, and includes sequence types ST23, ST26, ST57 and ST163¹⁴."

Q3. Line 116; To which STs did the 81 isolates belong to?

Response: The sequence types for the isolates included in this study are shown in Fig. 1 and the Supplementary Table 1.

Q4. Lines 234-42; without long read sequencing (that was done for the virulence plasmid), it is very difficult to judge the validity of AMR plasmid results. The authors should include a sentence stating that.

Response: Please see response to Reviewer 2 Q2; we have now resolved the equine AMR plasmid sequence.

Reviewer #1 (Remarks to the Author):

Overall, this revised manuscript is improved in clarity, and the authors have satisfied the majority of this reviewer's concerns. The reorganization to move additional experimental details to the supplemental text and revised description of results has made the manuscript easier to interpret and highlights the main takeaways of the analysis. To this reviewer, the main takeaways are the remote emergence of CG23, the detection of uncommon but possible carriage of antibiotic resistance plasmids, and the characterization of the subclade CG23-I that represents the majority of liver abscess isolates. My remaining concerns are that the virulence profile of CG23-I could be stated more clearly relative to other experimentally-characterized hypervirulent strains, and that the role of colibactin in virulence may be overstated relative to the experimental infections done by the authors and others.

Specific concerns:

In the discussion, the authors conclude that "the acquisition of the *clb* locus likely promotes both gut colonisation and mucosal invasion of CG23-I Kp, leading to enhanced transmissibility and increased virulence relative to ICEKp+ *clb*- strains...". This is based on the observation that colibactin is present in CG23-I but absent in other hypervirulent isolates (Figure 1). However, this statement is not tested experimentally, and other hypervirulent strains outside of CG23-I have been shown experimentally to efficiently colonize and invade. As the authors point out that experimental comparisons with other isolates is out of the scope of this work, it seems the conclusion above is too strong. Determination of the contribution of colibactin to colonization and virulence in this subclade can be tested in future work.

Describing the collection of validated and putative virulence genes present in CG23-I SGH10 compared to other well-characterized isolates is a main takeaway of the paper and would be more valuable to the field if stated more explicitly. This data is partially shown in Figure 1 and scattered throughout the discussion, but could be summarized more clearly. For example, the authors also discuss additional copies of *iro* and *rmpA* in NTUH-K2044 compared to other isolates but this is not represented in the phylogeny figures. A graphical summary of the shared and unique genes across the six reference isolates and SGH10 would enable direct testing of which genes are required and/or sufficient to enable hypervirulence when tested in isogenic mutants and across reference strains in animal models. This would support the authors' compelling conclusion that "no particular virulence factor of CG23 is necessary or sufficient for invasive disease" but the combination in CG23-I appears to have been most epidemiologically successful.

The phrase "with/without ICEKp +/- *clb* " within the discussion is confusing as written and makes the conclusion of the larger phase unclear.

Figure 3A: The bar in the scatterplot needs to be defined (mean, median).

Reviewer #2 (Remarks to the Author):

no further comments

Reviewer #3 (Remarks to the Author):

The authors addressed my concerns sufficiently.

Point-by-point response to remaining reviewer comments:

Q1. In the discussion, the authors conclude that “the acquisition of the *clb* locus likely promotes both gut colonisation and mucosal invasion of CG23-I *Kp*, leading to enhanced transmissibility and increased virulence relative to ICEKp+ *clb*- strains...”. This is based on the observation that colibactin is present in CG23-I but absent in other hypervirulent isolates (Figure 1). However, this statement is not tested experimentally, and other hypervirulent strains outside of CG23-I have been shown experimentally to efficiently colonize and invade. As the authors point out that experimental comparisons with other isolates is out of the scope of this work, it seems the conclusion above is too strong. Determination of the contribution of colibactin to colonization and virulence in this subclade can be tested in future work.

Response: We have amended the wording to “therefore, we speculate that acquisition of the *clb* locus promotes gut colonisation and/or mucosal invasion of CG23-I *Kp*, leading to enhanced...”. We agree that further work will be required to examine the overall contribution of colibactin alongside other virulence factors, as stated at the end of discussion paragraph 4.

Q2. Describing the collection of validated and putative virulence genes present in CG23-I SGH10 compared to other well-characterized isolates is a main takeaway of the paper and would be more valuable to the field if stated more explicitly. This data is partially shown in Figure 1 and scattered throughout the discussion, but could be summarized more clearly. For example, the authors also discuss additional copies of *iro* and *rmpA* in NTUH-K2044 compared to other isolates but this is not represented in the phylogeny figures. A graphical summary of the shared and unique genes across the six reference isolates and SGH10 would enable direct testing of which genes are required and/or sufficient to enable hypervirulence when tested in isogenic mutants and across reference strains in animal models. This would support the authors’ compelling conclusion that “no particular virulence factor of CG23 is necessary or sufficient for invasive disease” but the combination in CG23-I appears to have been most epidemiologically successful.

Response: We have further highlighted the differences in virulence loci by (i) adding columns to Figure 1 to indicate presence/absence of the chromosomally-encoded *iro* (salmochelin) and *rmpA* (regulator of mucoid phenotype) associated with ICEKp1, and (ii) adding a panel (b) to Figure 2 showing the presence and absence of the virulence loci against each of the complete CG23 genomes. These are highlighted in results text as follows:

SGH10 as a novel reference strain for hypervirulent CG23

Our data on CG23 population structure and virulence loci show that none of the currently available finished genome sequences represent a ‘typical’ genome (**Fig. 1 and 2b**). NTUH-K2044 (liver abscess), ED23 (blood) and NCTC 9494 (sputum) are not part of the predominant CG23-I sublineage and do not harbour the colibactin locus (labelled A-C in **Fig. 1**, note that NTUH-K2044 also carries additional chromosomal copies of *iro* and *rmpA* within ICEKp1 that are absent from the majority of CG23). Strains 1084 (liver abscess) and ED2 (blood) belong to CG23-I but lack the virulence plasmid, and RJF999 is a blood isolate of undetermined virulence.

Q3. The phrase “with/without ICEKp +/- *clb*” within the discussion is confusing as written and makes the conclusion of the larger phase unclear.

Response: Wording has been changed to “with or without yersiniabactin and/or colibactin”.

Q4. Figure 3A: The bar in the scatterplot needs to be defined (mean, median).

Response: The figure legend has been updated to specify that the bars indicate mean values.